# Beneficial Effects of Lactobacilli Species on Intestinal Homeostasis in Low-Grade Inflammation and Stress Rodent Models and Their Implication in the Modulation of the Adhesive Junctional Complex

**DOI:** 10.3390/biom13091295

**Published:** 2023-08-24

**Authors:** Célia Chamignon, Geoffroy Mallaret, Julie Rivière, Marthe Vilotte, Sead Chadi, Alejandra de Moreno de LeBlanc, Jean Guy LeBlanc, Frédéric Antonio Carvalho, Marco Pane, Pierre-Yves Mousset, Philippe Langella, Sophie Lafay, Luis G. Bermúdez-Humarán

**Affiliations:** 1Institut National de Recherche pour l’Agriculture et l’Environnement (INRAE), Micalis Institut, AgroParisTech, University of Paris-Saclay, 78350 Jouy-en-Josas, France; c.chamignon@novobiome.eu (C.C.); julie.riviere@inrae.fr (J.R.); sead.chadi@inrae.fr (S.C.); philippe.langella@inrae.fr (P.L.); 2INDIGO Therapeutics, 33000 Bordeaux, Francesophie.lafay@berkem.com (S.L.); 3INSERM U1107 NeuroDol, University of Clermont Auvergne, 63001 Clermont-Ferrand, France; geoffroymallaret@hotmail.fr (G.M.); frederic.carvalho@uca.fr (F.A.C.); 4INRAE, GABI, AgroParisTech, University of Paris-Saclay, 78350 Jouy-en-Josas, France; marthe.vilotte@inrae.fr; 5CERELA-CONICET, San Miguel de Tucumán T4000ILC, Tucumán, Argentina; demoreno@cerela.org.ar (A.d.M.d.L.); leblanc@cerela.org.ar (J.G.L.); 6Probiotical Research, 28100 Novara, Italy; m.pane@probiotical.com

**Keywords:** leaky gut, chronic diseases, barrier functions, probiotics, *Lactobacillus*

## Abstract

Intestinal barrier integrity is essential in order to maintain the homeostasis of mucosal functions and efficient defensive reactions against chemical and microbial challenges. An impairment of the intestinal barrier has been observed in several chronic diseases. The gut microbiota and its impact on intestinal homeostasis is well described and numerous studies suggest the ability of some probiotic strains to protect the intestinal epithelial integrity and host homeostasis. In this work, we aimed to assess the beneficial effects of three *Lactobacillus* strains (*Lacticaseibacillus rhamnosus* LR04, *Lacticaseibacillus casei* LC03, and *Lactiplantibacillus plantarum* CNCM I-4459) and their mechanism of action in low-grade inflammation or neonatal maternal separation models in mice. We compared the impact of these strains to that of the well-known probiotic *Lacticaseibacillus rhamnosus* GG. Our results demonstrated that the three strains have the potential to restore the barrier functions by (i) increasing mucus production, (ii) restoring normal permeability, and (iii) modulating colonic hypersensitivity. Moreover, gene expression analysis of junctional proteins revealed the implication of Claudin 2 and Cingulin in the mechanisms that underlie the interactions between the strains and the host. Taken together, our data suggest that LR04, CNCM I-4459, and LC03 restore the functions of an impaired intestinal barrier.

## 1. Introduction

The gastrointestinal tract (GIT) is the largest mucosal surface of the human body, which is in continuous contact with the external environment. The main functions of the GIT are the digestion of food and the absorption of essential nutrients and water. It also limits the infiltration of harmful substances, such as pathogens or toxins, into the intestinal mucosa to the blood circulation [1]. These functions are supported by intestinal barriers made up of the mucus layer, preventing the adhesion of large particles like bacteria, the epithelial cells layer, and the underlying *lamina propria*, containing immune components [2]. The selective permeability is provided by the paracellular pathway, which is mediated by the complex of junctional proteins that are, in turn, connected to the epithelial cells together [3]. The intestinal barriers represent a dynamic and responsive interface and play a crucial role in the maintenance of intestinal homeostasis and therefore healthy conditions of the host [4]. Moreover, the intestinal microbiota is now largely recognized as a key regulator of intestinal homeostasis by exerting beneficial effects at the genetic level [5] and maintaining general gut health [6].

The impairment of the epithelial barrier integrity is associated with a wide range of diseases related to the GIT as inflammatory bowel diseases (IBD), irritable bowel syndrome (IBS), or colorectal cancer, but also to non-gastrointestinal disorders such as obesity, diabetes, liver diseases, and neurodegenerative disorders [7]. Alterations of the epithelial barrier can lead to an increase in intestinal permeability (also known as “leaky gut”), promoting the passage of pro-inflammatory molecules and harmful microorganisms. In response to these stimuli, the immune system is activated, and chronic inflammation begins [8,9]. With a better understanding of epithelial homeostasis and disease pathogenesis, intestinal barrier functions seem to be a promising target [9].

Several strategies were proposed for the modulation of intestinal barriers, and the use of probiotic bacteria is continuously increasing. Indeed, the beneficial effects of probiotics are mediated by several mechanisms, which include the enhancement of the epithelial barrier that leads to the regulation of junctional protein expression, mucus production, and modulation of the immune responses and microbiota [10]. *Lactobacillus* species are largely consumed by humans, especially in dairy products, and also constitute the normal gut microbiota (especially in the small intestine). Probiotic lactobacilli demonstrated multiple beneficial effects for the regulation of intestinal homeostasis and are proposed as therapeutic agents for the prevention of disease development [11,12]. In this context, the *Lacticaseibacillus rhamnosus* GG strain (LRGG) is one of the best-studied probiotic bacteria, and its beneficial effects demonstrated from *in vitro* to clinical studies provide an interesting model of probiotic mechanisms [13]. However, *Lactobacillus* species do not exert the same effects on the host due to the differences in pathophysiological conditions and the multiple factors that modulate the intestinal barrier functions. It is thus important to determine the precise molecular mechanisms by which probiotic strains are able to restore intestinal homeostasis in order to provide targeted treatments of pathologies [14].

The aim of this work was to evaluate the beneficial effects provided by the selected *Lactobacillus* species (*Lacticaseibacillus rhamnosus* LR04 DSM 16605, *Lacticaseibacillus casei* LC03 DSM 27537, and *Lactiplantibacillus plantarum* CNCM I-4459, and the well-studied *L. rhamnosus* GG ATCC 53103) on intestinal homeostasis. Two models of low-grade inflammation and neonatal maternal separation (NMS) induced stress were used, and the inflammation, mucus production, paracellular permeability, and colonic hypersensitivity (CHS) were examined. Moreover, some of the mechanisms underlying the restoration of the barrier functions via the study of the complex of cellular junctions were also evaluated.

## 2. Material and Methods

### 2.1. Bacterial Culture Preparation

The origin, identity, and growth conditions of the strains that were used are shown in Table 1. Strains were stored at −80 °C in PBS 1X + 16% (*v*/*v*) glycerol.

For aliquot preparation of bacterial culture, bacteria were cultivated from a frozen stock of bacterial culture and were incubated overnight. This culture was passed once and incubated overnight. This second culture was centrifuged and washed with phosphate-buffered saline 1X (PBS) (Gibco, Carlsbad, CA, USA). The bacteria were then enumerated with FACS Accuri C6 cytometer (BD Biosciences, Franklin Lake, NJ, USA) and resuspended in PBS 1X + 16% glycerol solution in order to obtain a final concentration of 5 × 10^7^ CFU/mL.

### 2.2. In Vivo Models of Induced-Intestinal Hyperpermeability

Animals were housed in the Clermont-Ferrand animal facility; all experimentations were performed according to the ethical guidelines set out by the International Association for the Study of Pain (IASP), complied with the European Union regulation, and were approved by ethics committee C2EA-02 of Clermont-Ferrand (approvals CE110-12 and CE111-12).

#### 2.2.1. Low-Grade Inflammation

Each group was composed of 10 male mice C57BL/6J of 6 weeks old (Janvier Labs, Le Genest-Saint-Isle, France). The mice were acclimated for one week before experimentation inside the animal facility. The mice were anesthetized by an intraperitoneal injection of 0.1% ketamine and 0.06% xylazine. The inflammation leading to an increase in intestinal permeability was induced via intrarectal injection of a solution of 100 mg/kg of DNBS (Sigma Aldrich, Saint-Louis, MO, USA) and ethanol (EtOH) 30%. The group of mice without inflammation received the same intrarectal injection with a solution of EtOH 30% only. The mice were monitored throughout the experiment and particularly during the three days following the intrarectal injection. Ten days after this period, 200 μL of bacterial cultures at 1 × 10^7^ CFU (see bacterial culture preparation) were administered to the mice by oral forced feeding, every day for 10 days. Control groups with and without inflammation received 200 μL of PBS 1X + 16% glycerol solution (Figure 1). Groups were dispatched as follows: control without inflammation (PBS/EtOH), control with inflammation (PBS/DNBS), LR04 treatment (LR04/DNBS), LrGG treatment (LrGG/DNBS), and LC03 treatment (LC03/DNBS). The inflammation was reactivated 3 days before the end of the experiment with a second intrarectal injection of 100 mg/kg of DNBS.

#### 2.2.2. Neonatal Maternal Separation (NMS)

Stress was induced by neonatal maternal separation (NMS) for 3 h every day from day 2 until day 14. Pups were placed in individual boxes, in a separate room set up with similar environmental conditions. After this period, CHS induced by NMS was first assayed by performing CRD, (experimental procedure explained below). Only mice that displayed the highest responses to colorectal distension after NMS were used in this study. After NMS and CRD procedures, 200 μL of bacterial culture at 1 × 10^7^ UFC was administered to the mice via forced feeding, for a 10 day period (Figure 2). Each group was composed of 6 male C57BL/6J and 4 female C57BL/6J mice of 2 days old and were dispatched as follows: induced-hyperpermeability control (PBS/NMS), LrGG treatment (LrGG/NMS), LR04 treatment (LR04/NMS), CNCM I-4459 treatment (CNCM I-4459/NMS), and LC03 treatment (LC03/NMS). The PBS/NMS control group received 200 μL of PBS 1X + 16% glycerol solution.

### 2.3. Assessment of Tissue Damage

Mice were killed by cervical dislocation. The abdominal cavity was opened, and small and large intestines were removed. Colon samples were opened longitudinally, and macroscopic damages (ulcer, hyperemia, adhesion, and bowel thickness and length) were assessed immediately using calipers for the bowel thickness and a ruler for the length (Fisher Scientific, Waltham, MA, USA). The macroscopic score was attributed to colon samples using scoring systems published in McCafferty et al. study [15]. For histological damages, tissue samples were fixed with 10% formalin for 24 h, dehydrated with a gradient of EtOH and embedded in paraffin wax. Tissue samples were cut, using a microtome (Leica RM2255), into 5-µm-thick sections and mounted on adhesive microscope slides (Adhesives slides KP printer, KliniPath, Duiven, The Netherlands) and stained using hematoxylin, eosin, and saffron (HES). The colon and ileum tissues were visualized using a high-capacity digital slide scanner (Panoramic Scan 150, 3DHISTECH Ltd., Budapest, Hungary) and CaseCenter Viewer software (3DHISTECH Ltd., Budapest, Hungary). Microscopic findings were assessed quantitatively in arbitrary units using scoring systems described by Ameho et al. [16], and based on lymphatic infiltrations observations in colon tissues. For the small intestine, the length of five villi and the depth of five crypts were measured; the presence of infiltrative cells was also observed. The villus length/crypt depth ratio (VCR) was calculated to evaluate the functional status of ileum samples.

### 2.4. Quantification of Early Inflammation Biomarker: Lipocalin-2/NGAL (LCN2)

Frozen plasma samples were diluted (1/5000) in PBS 1X + 0.5% BSA (Sigma Aldrich, Saint-Louis, MO, USA) for the analysis. LCN2 levels were estimated using Duoset murine LCN2 Elisa Kit (R&D Systems, Minneapolis, MN, USA) as per the manufacturer’s instructions and expressed as ng/mL of plasma.

### 2.5. Study of Mucus Production

Mucin 2 (Muc2) proteins were labelled on formalin-fixed samples that were cut, using a microtome (Leica RM2255, Leica, Wetzlar, Germany), into 5-μm-thick sections and mounted on adhesive microscope slides (Adhesives slides KP printer, KliniPath, Duiven, The Netherlands). Immunostaining was performed using the Leica Bond RXm. Heat-induced antigen retrieval was done with Leica Bond™ Epitope Retrieval 1 (pH6) for 20 min. Rabbit polyclonal antibody against the MUC2 protein (NBP1-31231, Novus Biologicals, Littleton, CO, USA), diluted at 1/500, was revealed by a goat anti-rabbit Alexafluor 568 IgG (dilution 1/2000, Invitrogen, Waltham, MA, USA) with 4′,6-diamidine-2′-phenylindole dihydrochloride (DAPI, Invitrogen, Waltham, MA, USA) to counterstained nuclear. All the reagents were diluted with Bond Antibody Diluent (AR9352, Leica, Wetzlar, Germany). Slides were mounted using a fluorescent mounting medium (Fluoromount-G, Clinisciences, Nanterre, France), scanned with the Panoramic Scan 150 (3D Histech, Budapest, Hungary) and analyzed with the CaseCenter 2.9 viewer (3D Histech, Budapest, Hungary). Ten crypt (colon) or villus/crypt (ileum) units were analyzed in each sample, and the number of Muc2 cells was counted according to Fukuda et al. [17]. Tissue samples with non-well-oriented crypts were removed from the analysis.

### 2.6. In Vivo Permeability Assay

For both models, intestinal permeability was evaluated before and after the bacterial treatment period, via the use of a fluorescent molecule FITC-dextran (4 KDa) (TdB Labs, Uppsala, Sweden). Mice were forced-fed with 200 μL of 0.6 mg/g of body weight FITC-dextran solution. After 3.5 h of FITC administration, a blood sample was collected by retro-orbital sampling. Samples were collected and the fluorescence intensity was measured using a fluorometric plate reader (Tecan, Männedorf, Switzerland) at 488 nm excitation and 520 nm emission. The FITC labelled-dextran concentration in mice plasma was determined against standard concentrations of FITC.

### 2.7. Evaluation of the CHS

On the experimental day, mice were acclimated to restriction cages and tape-maintained on the tail for 1 h prior to colorectal distension (CRD), a non-invasive manometric method described by Larauche et al. [18]. Briefly, mice were anesthetized with 2.5% isoflurane in order to insert a miniaturized pressure transducer catheter (Mikro-Tip SPR-254; Millar Instruments, Houston, TX, USA) equipped with a custom-made balloon of 1.5 cm length. To avoid any colonic compliance effect, a balloon was prepared from a polyethylene plastic bag and inserted at approximately 10 mm from the anal margin. Mice were replaced in the holding device and allowed to recover 30 mn before CRD. The balloon was connected to an electronic barostat (Distender Series II, G&J Electronics, Toronto, ON, Canada) and a preamplifier (PCU-2000 Dual Channel Pressure Control Unit, Millar Instruments, Houston, TX, USA), in turn, connected to PowerLab interface (AD Instruments, Dunedin, New Zealand). CRD consists of the application of intrarectal pressure (20, 40, 60, and 80 mmHg). IPV was recorded for 20 s, with a 4 min interval between applied pressure, and analyzed using LabChart version 7 software.

### 2.8. Study of the Expression of Tight Junction Complex by Quantitative Real-Time PCR (qPCR)

The total RNA was isolated from intestinal tissues (duodenum, jejunum, ileum, distal proximal and median colon) generated after *in vivo* experiments using RNeasy Mini Kit (Qiagen, Hilden, Germany). On-column rDNase digestion was made to remove the residual content of DNA. RNA quantity and quality were checked with the NanoDrop apparatus (ThermoFisher Scientific, Waltham, MA, USA). RNA was used for subsequent cDNA synthesis with a High-Capacity cDNA Reverse Transcription Kit (ThermoFisher Scientific, Waltham, MA, USA), and 1 μg of the total RNA preparation was used for each sample. Quantitative real-time PCR (qPCR) was performed with diluted cDNA (100×) with StepOnePlus Real-Time PCR (Applied Biosystems, Waltham, MA, USA). The reaction mix was composed of Taqman Gene Expression Master Mix (see Table 2 for each primer tested) (ThermoFisher Scientific, Waltham, MA, USA), Taqman Gene Expression assay (ThermoFisher Scientific, Waltham, MA, USA) at 1X and 5 μL of diluted cDNA. Values are expressed as relative fold change normalized with the housekeeping genes, TBP, ACTB, and RPL19, by the 2^−ΔΔCT^ method. The running method was set up according to the manufacturer’s instructions.

### 2.9. Statistics

Results are expressed as the mean and standard error mean (SEM) and analysis was performed in GraphPad Prism version 8 (GraphPad Software, La Jolla, CA, USA). We carried out ordinary two-way ANOVA, followed by Dunnet’s multiple comparison test. For non-parametric data sets, scores, or percentages, Kruskal–Wallis tests, followed by Dunn’s multiple comparison test were performed. *p*-values below 0.05 were considered significant.

## 3. Results

### 3.1. Assessment of Tissue Damage

#### 3.1.1. Low-Grade Inflammation

The dinitrobenzene sulfonic acid (DNBS) model involved two intrarectal injections separated by a 10-day recovery period. We wanted to confirm the absence of major damage to the colon mucosa via the measurement of histological scores. As shown in Figure 3A, minor changes were observed in the mucosal surface of colon tissues. We further explored the histological changes and inflammation state in the ileum and colon samples by measuring the villus length/crypt ratio (VCR) and histological scores (Figure 3B). Concerning the ileum samples, no reduction of VCR was observed after the DNBS instillation; however, a slight increase was observed with CNCM I-4459 and LrGG treatments. Globally, the colon and ileum samples presented normal to blunting architecture with infiltration of mononuclear cells in the *lamina propria* (Figure 3C).

#### 3.1.2. Neonatal Maternal Separation (NMS)

In this model, no technical manipulation was carried out on mice, in order to explore the impact of stress on histological findings. As expected, we did not observe any change in the mucosal surface of colon tissues (Figure 4A). From histological findings, most of the colon and ileum samples presented normal architecture with some infiltration of mononuclear cells in the *lamina propria* (Figure 4C). While no change was observed in VCR and histological scores in response to CNCM I-4459 and LR04 strains, a slight reduction of the microscopic findings in the colon was observed for both treatments in colon tissues (Figure 4B).

### 3.2. Evaluation of Early Inflammation (Lipocalin-2)

We then evaluated the systemic inflammation commonly found to be associated with the induction of hyperpermeability, in both models (DNBS and NMS). For this, the early biomarker lipocalin-2 (LCN-2) was quantified in mice plasma. In both models, we were able to detect LCN-2 in the plasma samples collected directly at the end of the experiments. However, no significant increase in LCN-2 was observed between groups. LCN-2 concentration was almost 3 times higher in the DNBS model (599 ng/mL ± 71, Figure 5A) than in the NMS experiment (258 ng/mL ± 52, Figure 5B), in regard to control groups (PBS/DNBS and PBS/NMS). A significant decrease in LCN-2 concentration was observed with LC03 and LR04 treatments (*p* < 0.05).

### 3.3. Lactobacillus Strains Enhance the Epithelial Barrier Functions

#### 3.3.1. Effects on Mucus Production

The four *Lactobacillus* strains were evaluated for their effects on mucus production by quantification of the mucin 2 (Muc2) protein in intestinal tissues (ileum and colon). In the DNBS model, no significant change was observed in Muc2 expression between the control groups. We further explored the samples by scoring Muc2 protein expression in the crypt (colon) or villus/crypt unit (ileum) (Figure 6A). We did not observe differences in Muc2 expression in ileum tissues (immunofluorescence images are not shown); however, a significant increase in Muc2 cells in colon tissues was observed with LrGG treatment. In NMS mice, significant effects with the bacterial treatments were observed in both small and large intestines samples. The treatment with LC03 strain demonstrated a beneficial effect on Muc2 expression in colon and ileum samples. On the other hand, LrGG and CNCM I-4459 strains displayed a significant increase in Muc2 cells in ileum tissues, while the most significant effects were observed with LR04 treatment, in colon tissue (Figure 6B).

#### 3.3.2. Modulation of the Gut Permeability

The beneficial effects of the selected *Lactobacillus* strains were assessed on intestinal permeability using the paracellular fluorescent tracer, FITC labelled-dextran. Mice that were treated with DNBS presented a significant increase in the permeability to FITC molecules (*p* < 0.05) that validate the alteration of the intestinal barrier (Figure 7A). In this model, most of the strains showed a significant decrease in intestinal permeability (*p* < 0.05). In contrast, FITC concentration in mice plasma treated with CNCM I-4459 was similar to the one of the PBS/DNBS control group. For the NMS model, an overall reduction of intestinal permeability was observed with each bacterial group tested (Figure 7B). A major effect of LR04 and LC03 treatments (*p* < 0.05) was demonstrated, in comparison to maternal separated mice group.

#### 3.3.3. Modulation of the CHS

The beneficial impact of *Lactobacillus* treatments on the CHS was evaluated by colorectal distension (CRD) experiment. Intracolonic pressure variations (IPV) were significantly higher at the maximum distension pressure (80 mmHg), but also at a lower one (60 mmHg), in the control group of mice that received DNBS injection and compared to the healthy control group (PBS/EtOH) (*p* < 0.05). LC03 and LR04 strains evaluated in the DNBS model demonstrated a significant decrease in IPV, at maximum distension pressure, which indicates their beneficial effect on the CHS. We also observed a tendency of CNCM I-4459 treatment to decrease IPV. However, IPV observed with the LrGG strain was similar to the PBS/DNBS group (Figure 8A). In the NMS model, the three *Lactobacillus* strains tested showed an effect on the CHS by displaying a significant diminution of IPV variations at the maximum pressure, but also at lower pressures with LC03 (40 and 60 mmHg) and CNCM I-4459 (60 mmHg) treatments (*p* < 0.05). In contrast to the DNBS model, LrGG treatment presented a mild decrease in IPV, but was not significant (Figure 8B).

### 3.4. Expression of Gene Involved in the Complex of Epithelial Cell Junctions

To better understand the effects of *Lactobacillus* strains on the barrier functions, the expression of 13 relevant genes of the complex of junctional proteins was measured via RT-qPCR. In addition, each part of the gastrointestinal tract (duodenum, jejunum, ileum, ascendant, median, and descendant colons) was analyzed in order to determine their specific target throughout the gut. These analyses were realized in both the DNBS and NMS models, and a vast set of data was generated; we thus focused on the modulations that were the most significant. In both models, the Claudin 2 (CLDN2) protein gene was overexpressed in groups of mice treated with LR04, LC03, and CNCM I-4459 strains, and the most significative effects were observed in the ileum and ascendant colon (*p* < 0.05). In the DNBS model, the overexpression of CLDN2 was less significant in the median part of the colon (not even significant for LR04 treatment) in comparison to the NMS model. The control groups and LrGG-treated mice displayed lower expressions of CLDN2 in both models (Figure 9A). This overexpression was not observed with LR04, LC03, and CNCM I-4459 strains in the distal colon of the NMS mice, and the CLDN2 gene was even not expressed in LrGG-treated mice (Figure 9B). In addition to these observations, an increase in the expression of the Cingulin (CGN) gene was noted with LC03 and CNCM I-4459 in DNBS-treated mice (Figure 9A). The modulation of CGN expression, in the DNBS model, was particularly associated with CLDN2 overexpression, with the most significant effects observed with LC03 treatment in the ileum, ascendant colon, and median colon samples (*p* < 0.05). On the other hand, the CGN gene in NMS-treated mice was the most differentially expressed in the jejunum, ileum, and descendant colons with the LC03 strain (Figure 9B). In regard to the CNCM I-4459 strain, the results showed a significant modulation of the CGN gene in the jejunum and ileum for the DNBS model, while the CGN gene was modulated in the median part of the colon for the NMS model (*p* < 0.05).

## 4. Discussion

A variety of pathological conditions are associated with abnormal intestinal permeability, which is critical for the development or maintaining intestinal integrity and for preventing extra-intestinal diseases [19,20,21]. Multiple environmental factors influence intestinal integrity, including dietary lifestyle [22], drugs and antibiotics administration [23,24], pathogens expansion and chronic stress [25,26]. In this study, four different *Lactobacillus* strains, LR04, CNCM I-4459, and LC03 were evaluated for their effects on the alterations of the intestinal interface, induced via either intrarectally DNBS injection or chronic stress. Chemically induced colitis with DNBS agent, is a commonly used murine model that recreates the morphological, histopathological, and clinical characteristics of chronic intestinal disorders. Moreover, this model is largely used to test the beneficial effects of probiotics on intestinal barrier functions [27]. Low DNBS concentration and multiple injections were used to better approach the intestinal chronic disorders. NMS is a well-established model of early life stress, and it is known to induce intestinal alterations, largely conducted by the interactions between the gut and the central nervous system [28]. The low macroscopic values observed in both models demonstrate the non-invasive nature of hyperpermeability induction procedures. However, the macroscopic scores remain higher in the chemical induced-hyperpermeability model due to the direct injection of DNBS in the colon. The observation of inflammatory cell infiltration in the lamina propria of small and large intestine samples, from both animal models, indicates the presence of low-grade inflammation. We further explored chemically or stress-induced inflammation via quantification of LCN-2 in mice plasma. This protein expressed in normal tissues is highly increased in cases of inflammatory conditions as encountered in chronic diseases [29]. LCN-2 expression is also involved in several behavioral responses including emotional behavior, pain hypersensitivity, depression, and anxiety [30]. As expected, we observed a slight increase in LCN-2 concentration in the plasma of DNBS-treated mice, which validate the presence of low-grade inflammation. We also observed a significant decrease in LCN-2 concentration with CNCM I-4459 and LC03 treatments, which indicates the capacity of the strains to modulate the inflammation activated via DNBS injection. NMS mice expressed lower levels of LCN-2, in comparison with the DNBS model, which suggests the lower impact of stress on the development of low-grade inflammation. Moreover, no effect on LCN-2 concentration was observed with our bacterial strains, probably due to the low levels of LCN-2 released in the plasma of NMS mice.

The objective of this work was to determine the probiotical potential of selected *Lactobacillus* strains on the intestinal barrier functions, altered either by chemical or stress factors. As mentioned above, the alteration of the intestinal integrity promotes the passage of foreign elements including microorganisms and their toxins. The host responses are mediated by proinflammatory cytokines that are involved in the physiopathology of numerous chronic diseases, related or not to the GIT [31]. The intestinal mucus layer is not only a lubricator that facilitates the passage of fecal products, but it also provides the very first line of defense between the intestinal epithelium and luminal contents. It is the major source of nutrients for the commensals that live in the gut and fundamental for the host/microbiota interactions and intestinal homeostasis [32]. The main constituents of intestinal mucus are Muc2 proteins that are secreted by goblet cells, throughout the whole intestine. Impaired Muc2 production was reported to cause an imbalance in the microenvironment of the intestinal mucosa, that involves host immune changes and the production of microbial metabolites [33]. The bacterial strains that were evaluated in this study demonstrated a significant upregulation of Muc2 expression and were differentially expressed in regard to the tissue sample and pathophysiological context. Interestingly, the LC03 strain seems to have the ability to modulate the Muc2 expression of NMS-treated mice in both ileum and colon tissues. However, we did not observe changes in Muc2 expression within the control groups, in both the upper and lower parts of the intestinal tract. It was previously reported that the mucus alterations are preferentially linked to Muc2 o-glycosylation, rather than Muc2 expression [34]. Moreover, the o-glycosylation of mucins is described to drive commensal bacteria by selection, trapping or exclusion of specific species [35]. It should be of interest to evaluate the modulation of *Lactobacillus* strains on the physical properties of mucus.

We further assessed the integrity of the gut barrier by analyzing the paracellular permeability with the fluorescent tracer FITC. As previously observed in chemically treated mice [36,37,38], the paracellular permeability was higher in the DNBS-treated control group. LC03, LR04, and LrGG strains displayed promiscuous effects in the restoration of normal intestinal permeability while CNCM I-4459 seems to have no effect. NMS is well known to affect physiological processes, including intestinal permeability [39,40]. In this regard, we hypothesized that hypersensitive mice displayed an intestinal hyperpermeability and our data suggest that LC03- and LR04-treated mice have the potential to restore the gut alterations, as observed in DNBS-treated mice. CNCM I-4459 and LrGG treatments demonstrated a slight tendency to restore normal permeability and displayed different effects in function to stress or inflammation.

The DNBS model is known to cause some physiological effects on the CHS, which is closely related to intestinal permeability since direct alteration of the barrier functions is induced in this model [41]. The stress induced by maternal separation influences neuronal and intestinal communications that affect the functionalities of the epithelial barrier and influence colonic pain [42]. In addition, peripheral factors, such as the intestinal microbiota and secreted metabolites, establish a cross-talk with the brain but also with the intestinal cells. In the context of stress, these interactions are altered and induce a persistence of the CHS [43]. We evaluated the anti-nociceptive effect of the *Lactobacillus* strains during CRD. Abdominal contractions or IPV represent a commonly used index of CHS associated with chronic abdominal pain in mice and is based on the measurement of IPV during the CRD [44]. In both models, the majority of the strains demonstrated their capacity to reduce the IPV, which indicates their relevance in the treatment of abdominal pain, which is a symptom commonly observed in intestinal chronic disorders. These results further highlight the association of intestinal permeability to CHS. It is interesting to note that the treatment with CNCM I-4459 strain was not efficient on the intestinal permeability of DNBS- and NMS-treated mice, whereas a beneficial effect was observed on IPV. These observations are not coherent with the previous ones. However, they suggest that the bacterial strains should exert their beneficial effects on both central and peripheral dysregulations, within the gut-brain axis, to provide anti-nociceptive effects. We previously demonstrated the capacity of our strains to produce γ-aminobutyric acid, a major inhibitory neurotransmitter of the central nervous system, in the culture supernatants [45]. These findings provide new hypotheses in regard to their mechanisms of action and CHS.

*In vivo* and clinical studies demonstrated the role of cellular junctional proteins in paracellular permeability and clinical manifestations of abdominal pain [46,47]. Indeed, this complex of junctions is essential for barrier formation and the selective regulation of paracellular permeability. In addition, these proteins were described to regulate junctional assembly, cell polarization, gene expression, and cell proliferation [48]. We wanted to better understand the mechanisms by which *Lactobacillus* strains were able to modulate the barrier functions. For that, we chose 13 genes that code for the proteins of the cellular complex of junctions, and we evaluated their expression in regard to the pathophysiological context, bacterial strain treatments, and specific areas of the intestinal tract. The CLDN2 gene was overexpressed in the DNBS and NMS models in all segments of the small and large intestines, with LR04-, LC03-, and CNCM I-4459-treated mice. These observations are in contradiction with the literature since CLDN2 overexpression was originally reported in pathological states such as ulcerative colitis or IBD [49,50]. CLDN2 is a transmembrane protein located at the apical side of epithelial cells and is reported as pore-forming proteins that allow the selective transport of small cations and water molecules; fluxes of large and uncharged substances, such as 4 kD dextran, are unchanged [51,52]. Nowadays, the role of CLDN2 in intestinal homeostasis is controversial and the mechanisms that underlie the changes in CLDN2 expression are not well understood. Some *in vivo* studies demonstrated the potential ability of CLDN2 to impact cellular processes, such as apoptosis and tissue regeneration, via the modulation of cell proliferation and inflammatory pathways [53,54]. Moreover, CLDN2 highly interacts with a variety of signaling and regulatory proteins, which are contained in the cytoplasmic plaque and, in turn, linked to actin and keratin filaments, which are responsible for the cell structure and shape [55]. Among the cytoplasmic proteins, CGN seems to be involved in the tight junction assembly and is described to regulate CLDN2 gene expression [56,57]. Interestingly, CGN gene expression was increased with LR04-, LC03-, and CNCM I-4459-treated mice and these observations were particularly relevant in the DNBS model, where the most significant CLDN2 overexpression was in concordance with CGN increase, in the ileum and ascendant colon tissues. Some studies revealed the signaling pathways that underlie these interactions and suggested the role of RhoA activity [58,59,60], GEF, and GATA-4 signaling molecules [61,62]. These pathways seem to play a key role in cell proliferation and differentiation; however, evidence collected to date remain unclear due to the wide signaling network that conducts the modulation of junctional proteins. Many of these pathways are dysregulated during the disease state and protein expression is highly dependent on the study context. In our study, it is also important to take into consideration that these observations do not translate into either an increase in a functional protein or cellular localization and therefore require further investigations.

## 5. Conclusions

Taken together, our results demonstrate the probiotic potential of *Lactobacillus* species in two murine models of chronic low-grade inflammation and stress. In regard to the overall evidence, LC03 and LR04 seem to be the most promiscuous strains and we confirm their possible use in the treatment of intestinal chronic disorders such as IBS. In addition, we better described the underlying mechanisms by which *Lactobacillus strains* exert their beneficial effects and we found that the genes for CLDN2 and CGN proteins are implicated in these interactions. Finally, our study demonstrates strain-specific interactions with the host mucosa, differential targeted area of action throughout the GIT, and specific alteration of intestinal barrier functions in regard to the pathophysiological context. This study highlights the complexity of interactions and the importance to understand the mechanisms that impact the barrier functions and the interactions between the bacterial strain of interest and the host, in order to provide effective therapies for specific diseases.

## Figures and Tables

**Figure 1 biomolecules-13-01295-f001:**
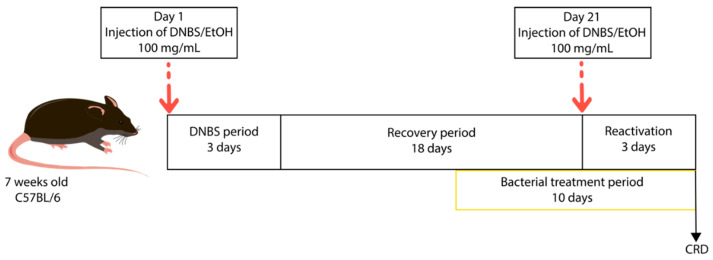
Schematic representation of the chronic low-grade inflammation model. Mice were inflamed via intrarectal administration of 100 mg/kg of DNBS solution in 30% EtOH. Control mice (without inflammation) received an equivalent amount of 30% EtOH.

**Figure 2 biomolecules-13-01295-f002:**
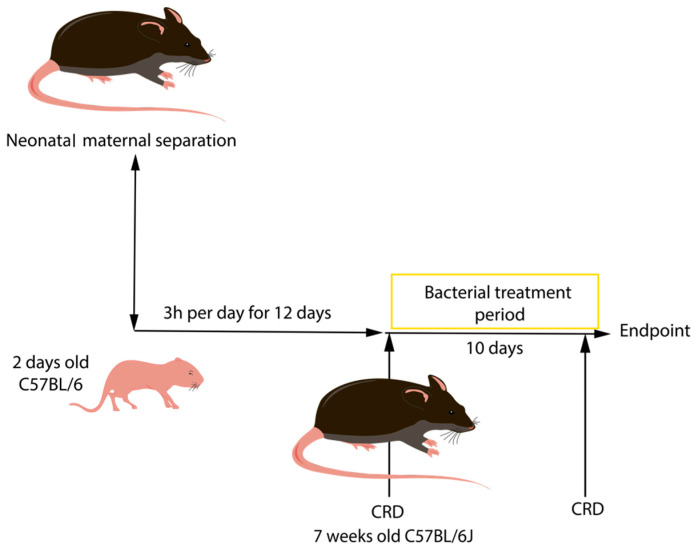
Schematic representation of the neonatal maternal separation model. Intestinal hyperpermeability was induced by the separation of the mice and the pups for 3 h per day for 12 days. CRD measurements were performed on mice before the bacterial treatment period to only obtain hypersensitive animals.

**Figure 3 biomolecules-13-01295-f003:**
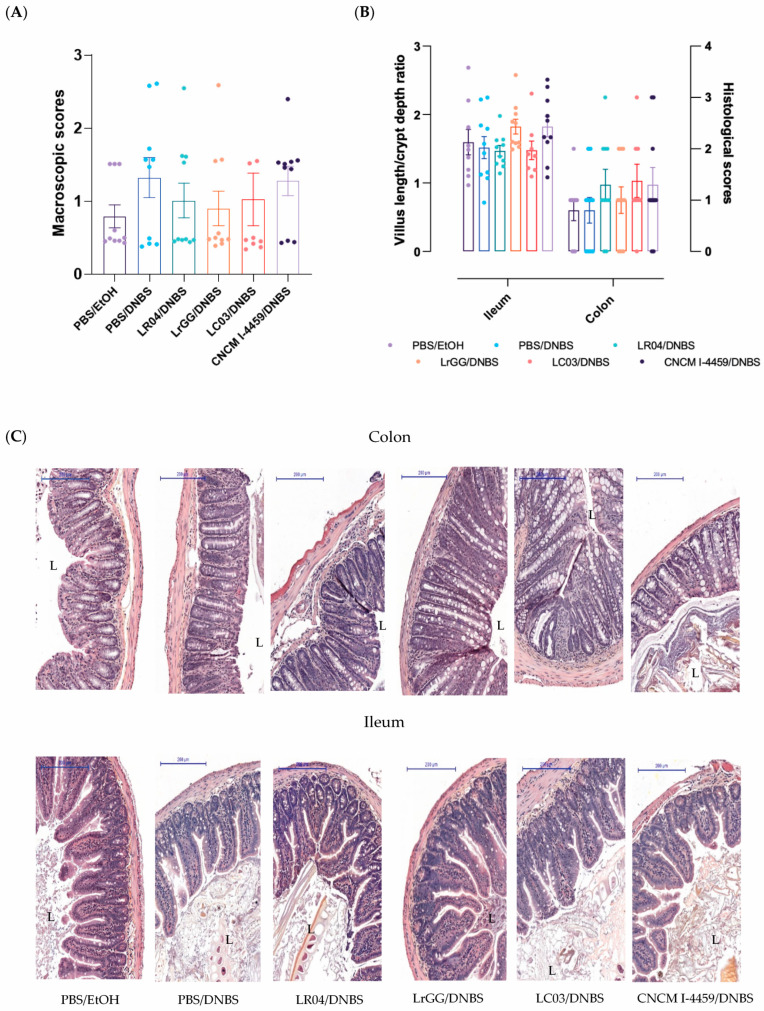
Assessment of tissue damage in DNBS induced-hyperpermeability model. The severity of the reactivated inflammation was assessed by macroscopic arbitrary unit analysis in the colon (**A**) and microscopic analysis based on histological findings for colon tissue and villus length/crypt depth ratio (VCR) for ileum analysis (**B**). Representative microscopic pictures of colon and ileum appearance stained with haemotoxylin and eosin (HES), “L” indicates the luminal part of samples (**C**), Scale bar 200 µm. Six groups of mice were compared: normal permeability (PBS/EtOH), control induced-hyperpermeability (PBS/DNBS), LR04 (LR04/DNBS), CNCM I-4459 (CNCM I-4459/DNBS), LrGG (LrGG/DNBS), and LC03 (LC03/DNBS). Statistical analysis was performed using the non-parametric Kruskall–Wallis test, followed by Dunn’s multiple comparison test. Non-significant results are not shown.

**Figure 4 biomolecules-13-01295-f004:**
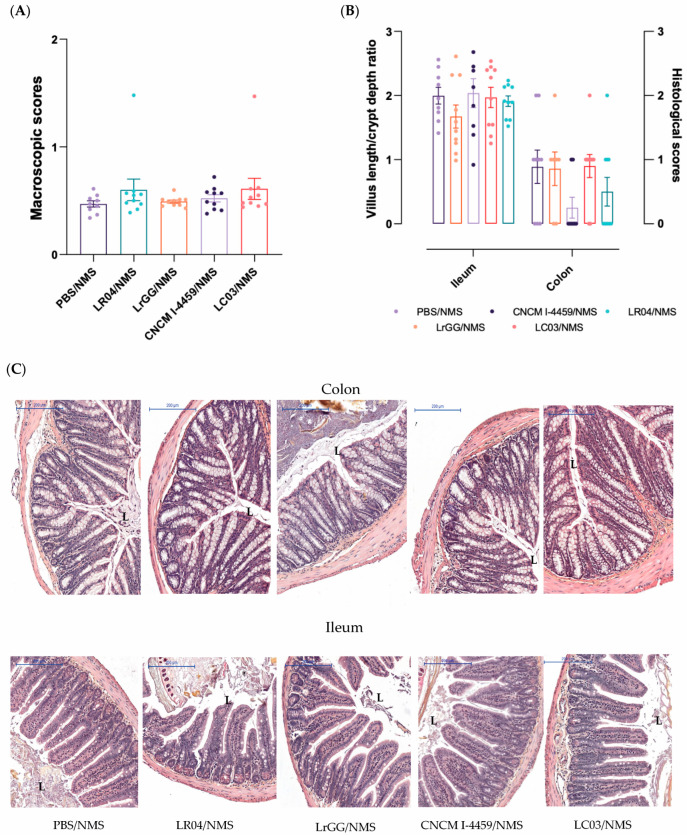
Assessment of tissue damage in stress-induced hyperpermeability. The impact of stress was assessed by macroscopic arbitrary unit analysis of colon tissue (**A**) and histological analysis based on lymphatic infiltration for colon tissue and villus length/crypt depth ratio (VCR) for ileum analysis (**B**). Representative microscopic pictures of colon and ileum appearance stained with haemotoxylin and eosin (HES) staining (**C**), Scale bar 200 µm. The different groups of mice were compared to control induced-hyperpermeability (PBS/NMS), LR04 (LR04/NMS), CNCM I-4459 (CNCM I-4459/NMS), LrGG (LrGG/NMS), and LC03 (LC03/NMS). Statistical analysis was performed using the non-parametric Kruskall–Wallis test, followed by Dunn’s multiple comparison test. Non-significant results are not shown.

**Figure 5 biomolecules-13-01295-f005:**
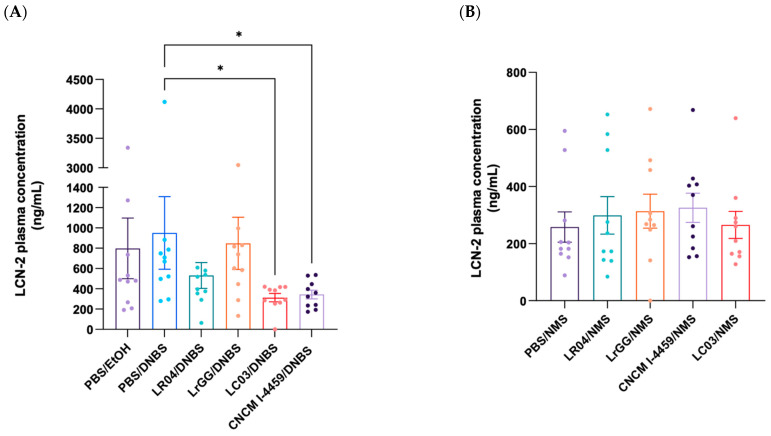
Effects of *Lactobacillus* strains on biomarker of early inflammation. LCN2 concentration was measured from mice plasma, collected after sacrifice in DNBS (**A**) and NMS (**B**) models. Results are presented as the concentration in ng/mL of LCN2 in plasma and compared with induced hyperpermeability controls (PBS/DNBS and PBS/NMS). Statistical analysis was performed using the non-parametric Kruskall–Wallis test, followed by Dunn’s multiple comparison test. * *p* < 0.0332. Non-significant results are not shown.

**Figure 6 biomolecules-13-01295-f006:**
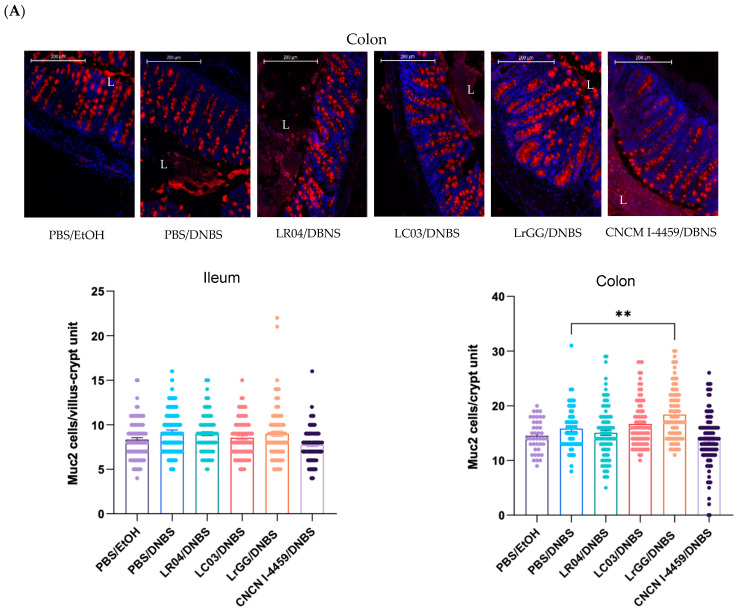
Capacity of the four *Lactobacillus* strains to modulate the production of mucus. Formalin-fixed samples were labelled using primary and secondary antibodies for Muc2 detection. The slides were mounted using a fluorescent mounting medium and were visualized with a high-capacity digital slider scanner and CaseCenter Viewer software. Representative images, Muc2 (red, Alexafluor 568) and nuclei (blue, DAPI), and associated scoring of Muc2 cells per villus/crypt (ileum) or crypt (colon) unit are shown for DNBS (**A**) and NMS (**B**) models, Scale bar 200 µm. Results were compared to induced-hyperpermeability controls (PBS/DNBS and PBS/NMS). Statistical analysis was performed using the non-parametric Kruskall–Wallis test, followed by Dunn’s multiple comparison test. **** *p* < 0.0001; ** *p* <0.0021; * *p* < 0.0332. Non-significant results are not shown.

**Figure 7 biomolecules-13-01295-f007:**
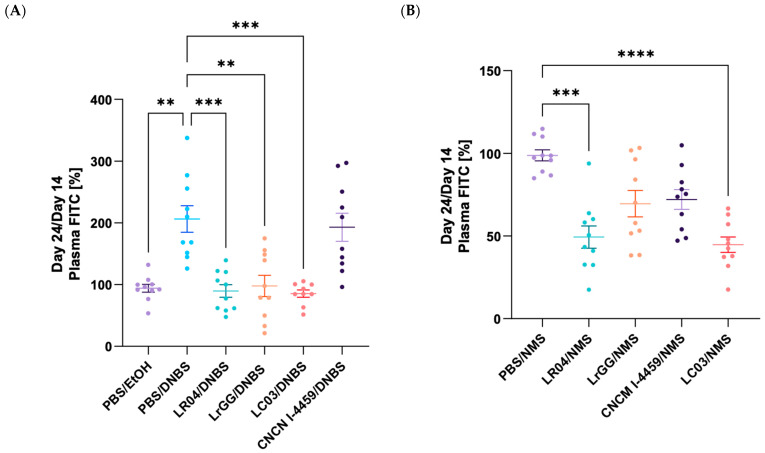
Effects of *Lactobacillus* strains on intestinal permeability. Mice were force-fed with FITC labelled-dextran solution of 0.6 mg/g of body weight. After 3.5 h, samples of blood were collected and the concentration of FITC-dextran in plasma was measured. (**A**) DNBS-induced hyperpermeability and (**B**) stress-induced hyperpermeability. Results are presented as the percentage of FITC in plasma and compared with induced-hyperpermeability controls (PBS/DNBS and PBS/NMS). Statistical analysis was performed using the Kruskal–Wallis test, followed by Dunn’s multiple comparison test **** *p* < 0.0001; *** *p* < 0.0002; ** *p* < 0.0021. Non-significant results are not shown.

**Figure 8 biomolecules-13-01295-f008:**
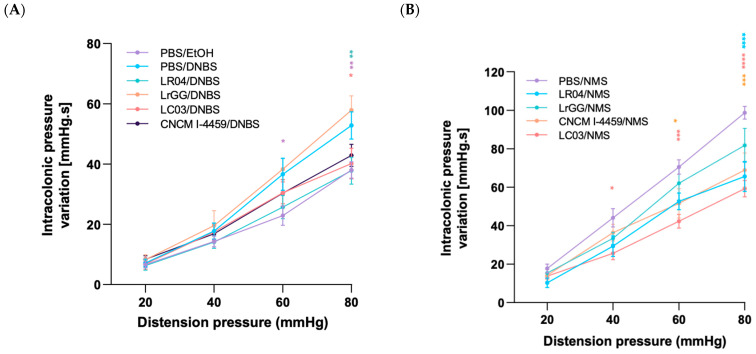
Effects of the four *Lactobacillus* strains on colorectal sensitivity. IPV of mice was evaluated in response to CDR, in DNBS (**A**) and NMS (**B**) models. Results are presented as intracolonic variation compared with induced hyperpermeability controls (PBS/DNBS and PBS/NMS). Statistical analysis was performed using the ordinary two-way ANOVA, followed by Dunnet’s multiple comparison test. **** *p* < 0.0001; *** *p* < 0.0002; ** *p* < 0.0021; * *p* < 0.0332. Non-significant results are not shown.

**Figure 9 biomolecules-13-01295-f009:**
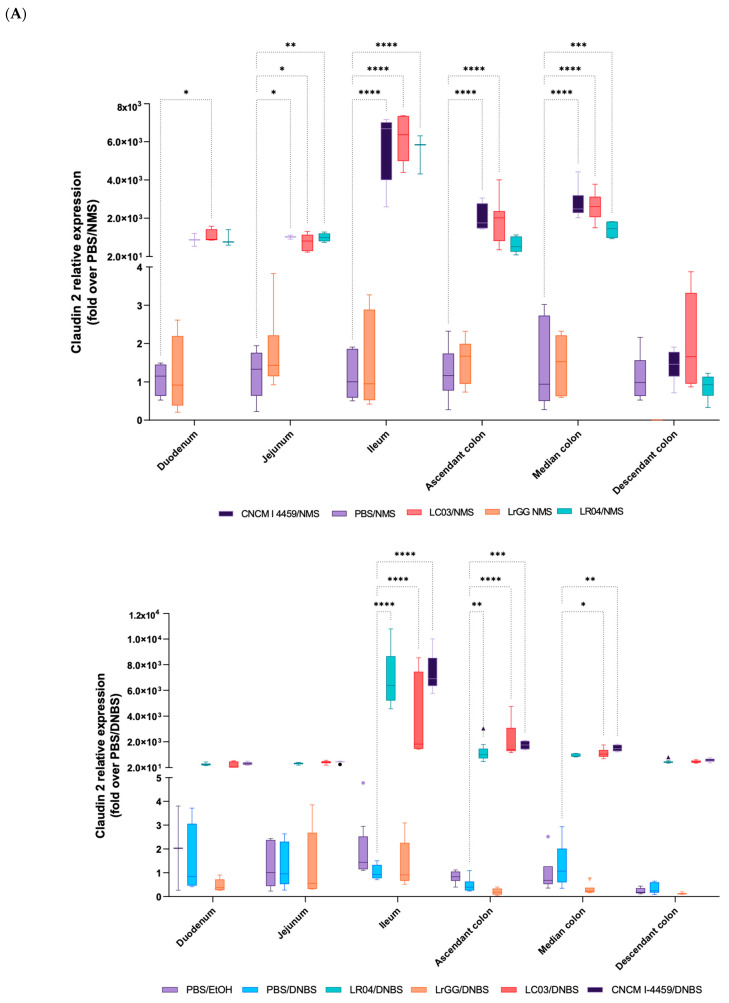
Modulation of junctional protein expression by *Lactobacillus* strains. Total RNA was isolated from intestinal tissues (duodenum, jejunum, ileum, ascendant colon, median colon and descendant colon) generated after *in vivo* experiments. RNA was used for subsequent cDNA synthesis. Quantitative real-time PCR was performed using Taqman reagents. Graphical data present the most relevant observations with relative expression of CLDN2 (**A**) and CGN proteins (**B**), in both experimental models of chemically or stress-induced hyperpermeability. Values are expressed as relative fold change normalized with the housekeeping genes, TBP, ACTB, and RPL19, via the 2^−ΔΔCT^ method. Statistical analysis was performed using the two-way ANOVA, followed by Dunnet’s multiple comparison test. **** *p* < 0.0001; *** *p* < 0.0002; ** *p* < 0.0021; * *p* < 0.0332. Non-significant results are not shown.

**Table 1 biomolecules-13-01295-t001:** Origin, identity and growth conditions.

*Lactobacillus* Species	Strains	Origin	Growth Conditions
*L. rhamnosus*	LrGG ATCC 53103	ATCC	Aerobic MRS 37 °C
*L. casei*	LC03 DSM 27537	Probiotical	
*L. rhamnosus*	LR04 DSM 16605	Probiotical	
*L. plantarum*	CNCM I-4459	CNCM	

**Table 2 biomolecules-13-01295-t002:** Genes used in this study.

Genes	Name	Assay ID
Claudin 1	CLDN1	Mm00516701_m1
Claudin 2	CLDN2	Mm00516703_s1
Claudin 3	CLDN3	Mm00515499_s1
Claudin 5	CLDN5	Mm00727012_s1
Occludin	OCLN	Mm00500912_m1
Junctional Adhesive Molecule (JAM)	F11-R	Mm00554113_m1
Cingulin	CGN	Mm01263534_m1
Zonula Occludens 1	TJP1	Mm00493699_m1
Zonula Occludens 2	TJP2	Mm00495620_m1
E-cadherin	CDH1	Mm01247357_m1
Vinculin	VCL	Mm00447745_m1
Desmoglein 2	DSG2	Mm00514608_m1
Myosin Chain Kinase	MYLK	Mm00653039_m1
Tata Binding Protein	TBP	Mm01277042_m1
Ribosomal Protein L19	RPL19	Mm02601633_g1
Actin Beta	ACTB	Mm02619580_g1

## Data Availability

The data published in this study are available on request from the corresponding author.

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
