# Peer review of "Beneficial Effects of Lactobacilli Species on Intestinal Homeostasis in Low-Grade Inflammation and Stress Rodent Models and Their Implication in the Modulation of the Adhesive Junctional Complex"

_biomolecules, 2023, doi:10.3390/biom13091295_

Round 1
Reviewer 1 Report
This manuscript addresses a research study that investigated the effects of three Lactobacillus strains as probiotics on inflammation, mucus production, intercellular permeability, and colon hypersensitivity (CHS) in a mouse model of low-grade inflammation or neonatal maternal isolation. The results of this study revealed that the three Lactobacillus strains restore barrier function by increasing mucus production, restoring intestinal permeability, and regulating colonic hypersensitivity. The topic addressed is interesting and deserves a constructive discussion. I think however that there are a few improvements that should be made before publication.
1. Is organic acid analysis in stools not measured before and after administration of the three types of Lactobacillus strains? If you have measured stool organic acids, please add that data.
2. The reason for using three types of Lactobacillus strains is unclear. Please explain a little more additionally why you selected those Lactobacillus strains.
Author Response
Reviewer #1:
Comments and Suggestions for Authors
This manuscript addresses a research study that investigated the effects of three Lactobacillus strains as probiotics on inflammation, mucus production, intercellular permeability, and colon hypersensitivity (CHS) in a mouse model of low-grade inflammation or neonatal maternal isolation. The results of this study revealed that the three Lactobacillus strains restore barrier function by increasing mucus production, restoring intestinal permeability, and regulating colonic hypersensitivity. The topic addressed is interesting and deserves a constructive discussion. I think however that there are a few improvements that should be made before publication.
- Is organic acid analysis in stools not measured before and after administration of the three types of Lactobacillus strains? If you have measured stool organic acids, please add that data.
Answer: Thank you for this question, however we have not measured the organic acids in the feces in this experiment.
- The reason for using three types of Lactobacillus strains is unclear. Please explain a little more additionally why you selected those Lactobacillus strains.
Answer: In a previous study (Chamignon et al., 2020: https://doi.org/10.3390/microorganisms8071053), we evaluated the probiotic properties and the beneficial effect on the cell membrane (transepithelial electrical resistance) of a panel of 50 strains. Thus, in this study, we decided to test the 3 strains of lactobacillus, the most interesting in our opinion, given their probiotic properties demonstrated in vitro in that study.
Reviewer 2 Report
General comments: The manuscript is written in a clear concise manner. I have very few comments (see below), but overall, this is a very well written manuscript. I recommend to accept after minor corrections.
Specific comments:
Line 16: In abstract, the opening word has mistakenly being made bold. Please make it normal font.
Line 81: The correct spelling is "Materials"
Lines 88-89: Replace "Phosphate Buffer Saline" with "phosphate buffer saline"
Author Response
Reviewer #2:
Comments and Suggestions for Authors
General comments: The manuscript is written in a clear concise manner. I have very few comments (see below), but overall, this is a very well written manuscript. I recommend to accept after minor corrections.
Specific comments:
- Line 16: In abstract, the opening word has mistakenly being made bold. Please make it normal font.
Answer: Thank you for your comment, we have made the corrections in the revised version of our Ms.
- Line 81: The correct spelling is "Materials"
Answer: Thank you, we have corrected this mistake.
- Lines 88-89: Replace "Phosphate Buffer Saline" with "phosphate buffer saline".
Answer: Done.
Reviewer 3 Report
The study "Beneficial effects of lactobacilli species on intestinal homeostasis in low-grade inflammation and stress rodent models and their implication in the modulation of the adhesive junctional complex" evaluate the beneficial effects provided by the selected Lactobacillus species on intestinal homeostasis
I found the study interesting, and the results are clear and consistent, I have few comments
1- The authors have to indicate significance or no significance on most of the graphs
2- Please consider these studies in your introduction and discussion
doi: 10.2147/JIR.S315938, and doi: 10.3390/ph14040341.
3- Label the miscroscopy channels on the photos of Figure 6
Moderate editing of English language required
Author Response
Reviewer #3:
Comments and Suggestions for Authors
The study "Beneficial effects of lactobacilli species on intestinal homeostasis in low-grade inflammation and stress rodent models and their implication in the modulation of the adhesive junctional complex" evaluate the beneficial effects provided by the selected Lactobacillus species on intestinal homeostasis
I found the study interesting, and the results are clear and consistent, I have few comments
1- The authors have to indicate significance or no significance on most of the graphs
Answer: Thanks for this precision, we have included now in the revised version of our Ms the non-significance of the corresponding results in the “figure legend” of the figure; "Non-significant results are not shown". We think it is better that the non-significant results are not annotated on the graph for easier reading.
2- Please consider these studies in your introduction and discussion
doi: 10.2147/JIR.S315938, and doi: 10.3390/ph14040341.
Answer: Thank you for this suggestion, we have added these references in the introduction and in the discussion.
3- Label the miscroscopy channels on the photos of Figure 6
Answer: Following this request, we have added the following phrase to the Ms: “Representative images, Muc2 (red, Alexafluor 568) and nuclei (blue, DAPI) and associated scor-ing of Muc2 cells per villus/crypt (ileum) or crypt (colon) unit are shown for DNBS (A) and NMS (B) models” where we indicated the dyes used, which also involves their respective excitation emission (i.e. channels).